# The Effects of Medications and the Roles of Pharmacists on the Recovery of Patients with COVID-19 Infection: An Epidemiological Study from the United Arab Emirates

**DOI:** 10.3390/healthcare11040467

**Published:** 2023-02-06

**Authors:** Iman A. Basheti, Hiba Barqawi, Razan I. Nassar, Samar Thiab, Noor Atatreh, Eman Abu-Gharbieh

**Affiliations:** 1Department of Clinical Pharmacy and Therapeutics, Faculty of Pharmacy, Applied Science Private University, Amman 11931, Jordan; 2Faculty of Medicine and Health, School of Pharmacy, The University of Sydney, Camperdown, NSW 2006, Australia; 3Research Institute of Medical and Health Sciences, University of Sharjah, Sharjah 27272, United Arab Emirates; 4Department of Clinical Sciences, College of Medicine, University of Sharjah, Sharjah 27272, United Arab Emirates; 5Department of Pharmaceutical Chemistry and Pharmacognosy, Faculty of Pharmacy, Applied Science Private University, Amman 11931, Jordan; 6College of Pharmacy, Al Ain University, Abu Dhabi 64141, United Arab Emirates; 7AAU Health and Biomedical Research Center, Al Ain University, Abu Dhabi 64141, United Arab Emirates

**Keywords:** coronavirus, COVID-19, pandemics, pharmacists, United Arab Emirates

## Abstract

Patients infected with coronavirus have new experiences and hence new needs from the healthcare sector. Acknowledging patients’ experiences can exhibit promising outcomes in coronavirus management. Pharmacists are considered a vital pillar in managing patients’ experiences during their infection. A cross-sectional study was conducted to assess the experiences of COVID-19-infected individuals and the roles of pharmacists in the United Arab Emirates. The survey was face- and content-validated after being developed. Three sections were included in the survey (demographics, experiences of infected individuals, and the roles of pharmacists). Data were analyzed using the Statistical Package for the Social Sciences. The study participants (*n* = 509) had a mean age of 34.50 (SD = 11.93). The most reported symptoms among participants were fatigue (81.5%), fever (76.8%), headache (76.6%), dry cough (74.1%), muscle or joint pain (70.7%), and sore throat (68.6%). Vitamin C was the most used supplement (88.6%), followed by pain relievers (78.2%). Female gender was the only factor associated with symptom severity. About 79.0% agreed that the pharmacist played an important and effective role during their infection. The most reported symptom was fatigue, with females reporting more severe symptoms. The role of the pharmacist proved to be vital during this pandemic.

## 1. Introduction

The first case of coronavirus disease (COVID-19) was reported in December 2019 in China, and since then, the disease has spread worldwide, causing more than 6 million deaths [1]. COVID-19 affects people differently; most infected people develop mild to moderate symptoms, such as fever, cough, fatigue, loss of taste or smell, and sometimes sore throat, headache, and diarrhea [2]. Some patients have more severe symptoms, such as shortness of breath or difficulty breathing, and thus require hospitalization [2,3].

The United Arab Emirates is a federation of seven emirates, with the largest population in Dubai (3,488,745), followed by Sharjah (1,786,000), Abu Dhabi (1,540,000), and Ajman (504,846) [4]. The first confirmed case of COVID-19 in the United Arab Emirates (UAE) was in January 2020 [5]. The UAE government adopted and implemented protective measures to control the spread of COVID-19 in March 2020, including closing schools and universities, along with switching to distance learning; closing shopping centers, entertainment destinations, and worship places; suspending flights; suspending issuing new visas; and switching to working from home, all in an effort to minimizes gatherings [3,6].

To date, there is no official treatment for COVID-19, but there are guidelines to help in choosing the most suitable medication to be used to relieve the symptoms of the infection [7]. As a result, people began taking over-the-counter (OTC) medications, supplements, and herbal products to boost their immunity and/or treat the disease [8]. Usually, such products are purchased from community pharmacies, and thus, pharmacists play a role in educating and managing COVID-19 infections, as well as continuing to care for patients with chronic diseases [9]. Several studies highlighted the importance of the pharmacists’ roles during the pandemic and how their efforts were well-perceived by the people in many countries, including Nigeria [10], Ghana [11], Syria [12], and China. As a result of these studies, it was concluded that the contribution of community pharmacists in the battle against the COVID-19 pandemic was a cornerstone in managing the disease burden on the health systems in low- and middle-income countries [13]. Furthermore, the International Pharmaceutical Federation (FIP) guidelines for pharmacists and the pharmacy workforce during COVID-19 highlighted that pharmacists could provide reliable information regarding COVID-19 detection and prevention [14].

A review was conducted to assess community pharmacists’ roles during the COVID-19 outbreak in the United States. It was found that in order to meet the rising demand for healthcare during COVID-19, pharmacists’ services rapidly expanded [15,16]. For example, they played a vital role in providing point-of-care testing, educating and encouraging the public to receive the COVID-19 vaccine [15]. Another review was conducted to assess the intervention and impact of pharmacist-delivered services in COVID-19 management. The COVID-19 pandemic led pharmacists to take on new clinical responsibilities, including the detection of COVID-19 and the avoidance of medication-related issues [17]. Pharmacists also played a crucial role in telemedicine during the COVID-19 outbreak, as they were providing medical care to patients, especially to those who were unable to leave their homes [18]. The COVID-19 infection continues to spread across the world, but the experiences of patients with the infection, symptoms, choice of medications, supplements, and herbs are not formally documented, and how pharmacists can improve that is still to be explored.

This study aimed to assess the experiences of COVID-19-infected individuals and to explore the roles of pharmacists in managing patients during their infection in the UAE.

## 2. Methods

### 2.1. Study Design and Subjects

This cross-sectional study was conducted in the UAE to address the study’s objectives. Data collection was carried out from the 4 of June 2022 to the 27 of July 2022 utilizing Google Forms. Participants were informed that their participation was voluntary and would not pose any risk to them. Participants who completed the study’s survey were considered by the research team to have given participation informed consent. 

### 2.2. Inclusion and Exclusion Criteria

The inclusion criteria included individuals who had lived in the UAE for at least a year, had been previously infected with COVID-19 at least once, and were above the age of 18. Exclusion criteria included individuals who did not comply with any inclusion criteria.

### 2.3. Survey Development, Validation, and Reliability

An extensive review of the literature was conducted to develop the first draft of the survey [19,20,21,22,23]. The research team created two surveys; the first was in English, and the second was in the Arabic language using the forward–backward translation method.

Five independent researchers who had previous experience in research and pharmacy practice reviewed the survey to provide the face and content validity of the survey. The suitability of the words, appropriateness and clarity of items, consistency of layout and style, item comprehension, and the relevancy of items to the objectives, as well as the study’s aim, were all evaluated by independent researchers. The research team took their feedback into account; thus, the items were changed and rewritten following their suggestions. They confirmed that the survey was clear, comprehensive, and free from medical jargon or complicated terminology.

The survey was then piloted on 50 participants who tested its applicability to enhance its understandability, readability, and clarity. The Cronbach alpha coefficient was used to assess internal consistency reliability and was found to be 0.86. The survey’s final version was divided into three primary sections; each section addressed a different area of interest. All the sections consisted only of closed-ended questions. The first section contained items to collect participants’ demographic information (age, gender, living place, marital status, medical insurance, smoking status, working in the medical sector, having a chronic disease, and having a respiratory disease). Clarification was added when appropriate; for example, chronic disease was defined as a condition that lasts for a year or longer, requires ongoing medical care, restricts daily activities, or both (e.g., hypertension and diabetes). Respiratory disease was defined as a condition that affects the lungs or the airways that impact human respiration, such as asthma and chronic obstructive pulmonary disease. The second section contained items to assess participants’ experiences with COVID-19 infection in general. In this section, participants were asked to rate their infection’s severity using a 5-point Likert scale, where 1 represented a non-severe infection, indicating that their daily life activities stayed normal, and 5 represented a very severe infection, indicating that their daily life activities were highly interrupted. In addition, participants were informed of the symptoms that would have affected their daily activities due to COVID-19 infection (shortness of breath, loss of appetite, confusion, persistent pain or pressure in the chest, and fever (temperature higher than 38 °C)). Section three contained seven questions to explore participants’ experiences with their pharmacists during their COVID-19 infection. The questions explored how effective the pharmacist’s role was in managing participants’ infection, whether they provided advice on how to control the viral infection and improve participants’ immunity, whether they provided instructions on how the medications were supposed to be used, clarified if any drug interactions were found, and provided warnings regarding medications that had to be avoided during the participant’s infection. Participants were also asked whether the pharmacists demonstrated the use of certain devices needed for COVID-19 infection management, such as the oximeter and the thermometer. In this last section, a 5-point Likert scale was used to document participants’ responses (5: strongly agree, 4: agree, 3: neutral, 2: disagree, and 1: strongly disagree). Results from the third section were presented as percentages of participants who showed each of the responses.

### 2.4. Survey Implementation

A link to the survey was distributed mainly via social media (WhatsApp and Facebook(. Participants were able to view the study’s ethics information and the study’s objectives before completing the survey. An average of 10 min was needed to complete the survey. This was clarified to the participants so they knew how much time was required to be allocated before completing the survey.

### 2.5. Sample Size

With a 5% margin of error, a 95% confidence level, and a 50% response distribution, the sample size was computed. The required minimum sample size was 385 participants [24,25,26]. To increase the study’s generalizability, efforts to contact a larger number of participants were made by the research team.

### 2.6. Statistical Analysis

After data collection, the Statistical Package for the Social Sciences (SPSS; Version 24.0; IBM Corp., Armonk, NY, USA) was used to analyze responses from participants. The mean (standard deviation) was used to represent descriptive data, while percentages were used to represent qualitative variables.

To determine the predictors of the dependent variable ‘infection severity’, a multiple linear regression analysis was performed after an assessment for collinearity. Associations between the dependent variables were analyzed against the independent variables: gender (1: Male, 2: Female), age, smoking status (1: Non-smoker, 2: Smoker), having a chronic disease (1: No, 2: Yes), and having a respiratory disease (1: No, 2: Yes).

## 3. Results

A total of 710 participants were recruited in the study; however, 201 out of the 710 participants reported that they were not infected previously with COVID-19; thus, they were excluded from the analysis, and consequently, a total of 509 participants were included, yielding a response rate of 71.7%.

The study participants’ (*n* = 509) had a mean age of 34.50 (SD = 11.93), and about 63.0% of the participants were females. Regarding participants’ living places, more than half of them (53.0%) were living in Abu Dhabi, 31.6% in Sharjah, and 10.4% in Dubai. More than half of the participants (55.6%) were married. The majority of participants had medical insurance (86.4%), were non-smokers (80.6%), did not work in the medical sector (88.6%), did not have any chronic diseases (88.6%), and did not have any respiratory diseases (94.5%). The demographic characteristics of the study participants are listed in Table 1.

### 3.1. Experiences of Infected Patients

Regarding the number of COVID-19 infections, more than half of the participants reported being infected once (51.9%), whereas 34.0% reported experiencing two infections, and 14.1% reported being infected more than two times. The time of infection varied widely among the participants; the highest percentage was reported in January 2022 (15.91%), followed by June 2022 (11.40%).

When participants were asked to rate their infection’s severity, more than one-third of them (37.3%) gave their infection a three out of five rating. Only 8.3% responded with a five, indicating that their infection was very severe. The infection severity mean was 2.85 (SD = 1.13).

About 73.0% of the participants stated that they received the COVID-19 vaccine before getting infected. Regarding the source of infection, 43.8% thought that COVID-19 was transmitted to them after contacting infected individuals who were family or friends, while 19.6% claimed their workplace was the source of their infection. Other sources were stated by the study participants, such as visiting a public place, for example, a restaurant or a shopping center, going to school or university, or traveling. On the other hand, 16.3% were unsure about their source of infection. A high percentage of the participants (96.5%) notified the people they contacted during the previous days regarding their COVID-19 infection.

Table 2 summarizes the percentage of experienced symptoms by the participants (*n* = 509). Fatigue was the most experienced symptom (81.5%), followed by fever (76.8%), headache (76.6%), dry cough (74.1%), muscle or joint pain (70.7%), and sore throat (68.6%).

Among those participants who experienced loss or change in taste or smell (54.8%, *n* = 279), 49.4% said it took less than a week for their symptoms to recover, 31.2% said it took between one week and one month, and 19.2% said it took longer than a month.

Prior to and following COVID-19 infection, participants were requested to rate their commitment to preventive measures, including wearing a mask and gloves (commitment rate was 4.27 (SD = 0.879)) and maintaining physical distance (commitment rate was 4.20 (SD = 0.983)). There was no statistically significant difference between the two commitment rates.

Participants reported that the health authorities, such as the Ministry of Health (MoH), Dubai Health Authority (DHA), and Abu-Dhabi Department of Health (DOH), contacted around 77.0% of the participants during their infection.

Around 30.0% of the participants needed to visit a doctor during their infection, 12.0% required hospitalization, and 2.8% needed an intensive care unit admission. A total of 23.0% of the study participants used an oxygen meter to measure their blood oxygen levels during their infection. More than half of the participants (55.0%) stated that their stress and anxiety levels increased during their COVID-19 infection.

About 60.0% of the participants (*n* = 308) reported using home remedies such as herbs, onions, cloves, and anise during their infection to relieve their symptoms. Moreover, among those participants (*n* = 308), 74.6% stated that these home remedies were helpful.

Table 3 shows the study participants’ percentage of used medications/supplements. Vitamin C was most commonly used (88.6%), followed by pain relievers, such as paracetamol (78.2%), zinc (56.4%), vitamin D (54.2%), and multivitamins (45.4%).

A total of 43.2% of the participants were advised to use these medications/supplements by a doctor, 17.1% were advised by a family member or a friend, 13.6% took the medications/supplements by themselves without consulting anyone, and 8.4% were advised by the pharmacist. On the other hand, 9.8% of the participants did not take any medication/supplement during their COVID-19 infection.

Around 30.0% of the study participants communicated with a pharmacist during their infection. Some of the participants contacted the pharmacist over the phone (47.5%), and others visited the pharmacy in person (24.8%) or through a third person (17.8%).

To unveil the predictors of the dependent variable ‘infection severity’, multiple linear regression analysis was performed and showed that participants’ gender (being female; *p* = 0.001) was positively associated with higher infection severity (Table 4).

### 3.2. Roles of the Pharmacists

Regarding the roles of the pharmacists during participants’ COVID-19 infection, 79.3% strongly agreed/agreed that “*The pharmacist had played an important and effective role during my infection*”, 74.4% strongly agreed/agreed that “*The pharmacist provided me with advice on how to control the viral infection*”, 76.6% strongly agreed/agreed that “*The pharmacist provided me with advice on how to improve my immunity*”, 84.5% strongly agreed/agreed that “*The pharmacist gave me instructions on how to use the medications*”, 64.1% strongly agreed/agreed that “*The pharmacist ensured that there were no drug interactions between my medications*”, 59.3% strongly agreed/agreed that “*The pharmacist warned me about medications to be avoided during my infection*”, and 55.9% strongly agreed/agreed that “*The pharmacist explained how to use some devices such as oximeter and the thermometer*”.

## 4. Discussion

This cross-sectional study provided insight into the experiences of people infected with COVID-19 in the UAE and highlighted the roles of pharmacists during the pandemic. The main study outcomes showed that more than half of the participants were infected at least once with COVID-19, and about three-quarters of them were already vaccinated. The most common symptoms among participants were fatigue, fever, headache, dry cough, muscle or joint pain, and sore throat. Sixty percent of the participants reported the use of home remedies, and the most commonly used vitamin C supplements and pain relievers. Female gender was the only factor associated with more severe symptoms.

According to the reported numbers by the WHO in 2022, the confirmed cases of people infected with COVID-19 were the highest during January; then, cases started to drop before spiking again in June, which is consistent with the experiences of the participants in this study [27]. As the UAE was among the countries that started the vaccination campaign early [28], most of the population were vaccinated by the time they participated in this study, and thus, around two-thirds of the participants were vaccinated before getting infected [29].

The symptoms that were reported by participants were consistent with those previously reported by the WHO [2] and were also reported by participants of a similar study conducted in Syria [12]. Additionally, fatigue, fever, and dyspnea were found to be very common among COVID-19 patients in two systematic reviews [30,31]. 

Loss of smell or taste is common in patients infected with COVID-19 [2]. In this study, around half of the participants reported experiencing this symptom and regained their sense of smell within a month, which is similar to the findings of a study conducted in Italy, where substantial improvements in olfactory senses were observed within 1–2 months of the onset of symptoms [32]. More than half of the participants reported higher anxiety levels when being infected, which was expected, as reported in different studies measuring anxiety levels in patients and families of COVID-19 patients [33,34,35,36,37].

Herbal remedies have gained popularity and were used to relieve COVID-19 symptoms [38,39,40,41]. Participants in the UAE demonstrated similar behavior, as more than half used these remedies and reported them to be useful. In a study conducted in Jordan, the use of citrus fruits, honey, and ginger was reported [8]. In another study conducted in Syria, ginger, clove, and anise were reported to be used to relieve the infection symptoms [12]. Additionally, over-the-counter medications and supplements such as vitamin C, vitamin D, and painkillers were used, as reported by participants in studies conducted in Jordan [8] and Syria [12], as well as in this study. Interestingly, the use of vitamins and supplements increased during the pandemic, even though there is limited therapeutic evidence of vitamins’ efficiency in treating COVID-19 [8]. However, several clinical reports highlighted the role of vitamins in fighting infections generally, including COVID-19 [42,43], which can explain the widespread use of these products in several countries around the world [44,45,46].

Doctors played a key role in the participants’ choice of supplements and medications, which is similar to the situation reported in other countries, including Syria [12]. Pharmacists were also contacted in different ways by the participants, and it was shown in various studies that community pharmacists are easily accessed [47,48,49] and have an active role in advising patients and correcting health-related misinformation [50]. An increase in the dependence on community pharmacists was observed worldwide during the pandemic because they were one of the first contact points to the public, providing medications, awareness, and advice [48,51].

In this study, only being of the female gender was associated with more severe symptoms during the infection period. Female gender was also found to be associated with higher severity of symptoms in a study conducted previously in Syria [12]. In a previously published literature review on the risk factors affecting COVID-19 severity, older age in females was found to be associated with higher severity of infection [52].

## 5. Study Strengths

This study was the first to explore patients’ experiences of their COVID-19 infection living in UAE, the medications they used, and their perceived assessment of the pharmacist’s role in managing their COVID-19 infection and providing counseling on medications and the use of needed devices, such as the oximeter. A step towards better patient-specific care is to emphasize the patient’s experience [12]. Managing COVID-19 infection in a suitable and effective manner can only be performed once patients’ own experiences are outlined. This study can add to the literature knowledge, which will result in better patient-specific care.

## 6. Limitations

The sampling method in this study depended on the availability of internet-based services to the participants and their willingness to participate in the survey, so people with limited internet access could not have been able to participate in this study. Additionally, more than half of the participants were females and from the capital, Abu Dhabi, which may not be representative of the whole population of the UAE. Furthermore, the study did not identify if participants were treated at home or hospitalized.

## 7. Conclusions

The majority of participants in this study were infected at least once with COVID-19, and more than half reported being infected more than twice. More than one-third of participants experienced average severity of symptoms, while about 8% experienced very severe symptoms, with females showing an association with higher severity. Two-thirds of participants received the COVID-19 vaccine, and 96.5% notified contacts during their infection, indicating that participants showed a high level of responsibility towards pandemic management. Health authorities contacted most participants during their infection, showing a high rate of follow-up of patients by their governments. With more than half of the participants stating that their stress and anxiety levels increased during their COVID-19 infection, mental health monitoring, and awareness campaigns would benefit people during pandemics. The majority of the participants used home remedies during their infection to relieve their symptoms, indicating the importance of healthcare professionals following up with COVID-19 patients, not only with their medication use, but with their herbal remedy usage as well. Pharmacists were found to have a vital role in managing patients during COVID-19 infection. They provided advice on how to control the viral infection and improve patients’ immunity and gave instructions on how to use medications and devices during the infection, such as oximeters and thermometers.

## Figures and Tables

**Table 1 healthcare-11-00467-t001:** Demographic characteristics of the study participants (*n* = 509).

Parameter	*n* (%)
Gender	
Female	321 (63.1)
Male	188 (36.9)
Living place	
Abu Dhabi	270 (53.0)
Sharjah	161 (31.6)
Dubai	53 (10.4)
Ajman	11 (2.2)
Ras Al-Khaimah	9 (1.8)
Fujairah	3 (0.6)
Umm Al-Quwain	2 (0.4)
Marital status	
Married	283 (55.6)
Single	210 (41.3)
Divorced	10 (2.0)
Widowed	6 (1.2)
Medical Insurance	
Yes	440 (86.4)
No	69 (13.6)
Smoker	
Yes	99 (19.4)
No	410 (80.6)
Working in a medical sector	
Yes	58 (11.4)
No	451 (88.6)
Having a chronic disease *	
Yes	58 (11.4)
No	451 (88.6)
Having a respiratory disease ^#^	
Yes	28 (5.5)
No	481 (94.5)

* Chronic diseases: a condition that lasts for a year or longer, requires ongoing medical care or restricts daily activities, or both (e.g., hypertension and diabetes). **^#^** Respiratory disease: a condition that affects the lungs or the airways that impact human respiration, such as asthma and chronic obstructive pulmonary disease.

**Table 2 healthcare-11-00467-t002:** The proportion of experienced symptoms reported by study subjects (*n* = 509) who previously developed COVID-19 infection.

The Symptom	*n* (%)
Fever	391 (76.8)
Shortness of breath	172 (33.8)
Dry cough	377 (74.1)
Muscle or joint pain	360 (70.7)
Nausea, vomiting, or diarrhea	159 (31.2)
Fatigue	415 (81.5)
Sore throat	349 (68.6)
Eye pain	109 (21.4)
Runny nose	290 (57.0)
Chills	233 (45.8)
Loss or change in the senses of taste or smell	279 (54.8)
Anorexia (loss of appetite)	209 (41.1)
Chest pressure or pain	132 (25.9)
Pneumonia	50 (9.8)
Headache	390 (76.6)
Hair loss	129 (25.3)

**Table 3 healthcare-11-00467-t003:** The medication/supplement used by the study participants (*n* = 509) during their COVID-19 infection.

The Medication/Supplement	*n* (%)
Vitamin C	451 (88.6)
Vitamin D	276 (54.2)
Zinc	287 (56.4)
Multivitamins	231 (45.4)
Cortisone	27 (5.3)
Pain relievers such as Paracetamol	398 (78.2)
Cough syrup	222 (43.6)
Antibiotics such as Azithromycin *	157 (30.8)
Blood thinners such as Aspirin (acetylsalicylic acid) *	39 (7.7)
Antacids such as Nexium (Esomeprazole) and Lanzotec (Lansoprazole) *	42 (8.3)
Antivirals medicines such as Favipiravir *	55 (10.8)

* Antibiotics: a medicine that inhibits or slows down the growth of microorganisms. Blood thinner: a medicine that prevents or lessens the coagulation of blood. Antacid: a medicine which neutralizes stomach acidity. Antivirals: a medicine that kills a virus or inhibits its ability to replicate, preventing it from multiplying and reproducing.

**Table 4 healthcare-11-00467-t004:** Summary of the regression model obtained for the dependent variable ‘infection severity’ (*n* = 509).

Variable	Infection Severity Dependent Variable		
	Beta	*t*	*p*-Value
Gender	0.164	3.432	0.001
Age	−0.062	−1.344	0.179
Smoking status	0.005	0.100	0.920
Having a chronic disease	0.036	0.779	0.436
Having a respiratory disease	0.073	1.640	0.102

This table shows the output from a multivariable regression analysis in which the severity of infection was the dependent variable. The backward regression method was used. The overall fit of the model was R^2^ = 0.039, *p* = 0.001. “Beta” is the standardized regression coefficient. Numbers in “bold” indicate significant results.

## Data Availability

The data presented in this study are available on request from the corresponding author. The data are not publicly available due to privacy.

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
