# Peer review of "The Effects of Medications and the Roles of Pharmacists on the Recovery of Patients with COVID-19 Infection: An Epidemiological Study from the United Arab Emirates"

_healthcare, 2023, doi:10.3390/healthcare11040467_

Round 1
Reviewer 1 Report
1-The title you have chosen is not appropriate, it appears to be a descriptive and epidemiological study and only the effect of pharmaceuticals on the recovery of covid-19 has been examined.
2-In the methodology, it is mentioned that in your survey have "a 5-point Likert scale", but this criterion is not described clearly.
3-The variable of severity of infection is not defined also "chronic disease" in table 1 and Antibiotic in table 3.
4-The criteria for inclusion and exclusion from the study have not been defined.
5-Tables have not been used to express data clearly and appropriately, and only percentages and descriptions have been used.
6-Were the subjects of the study hospitalized or treated at home?
Author Response
I appreciate the suggestions, which have been very helpful in improving the manuscript. I also thank you for your careful reading of the manuscript
All the received comments on this study have been taken into account in improving the quality of the article, and I present below the reply to each of them separately.

Reviewer 2 Report
Dear Authors, I have some suggestions to improve it further.
1. The introduction lacks a review of Pharmacist role in COVID-19 care. Please read these manuscripts and cite them. Also search more literature.
https://www.mdpi.com/2227-9032/10/9/1630
https://journals.sagepub.com/doi/pdf/10.1177/0897190020980626
2. Methods, Inclusion and exclusion criteria is not clear. Write it in separate heading.
3. Methods, questionnaire validation process results are missing. Did you done pilot study? Chronbach alpha? Who reviewed the questionnaire?
4. Whole method section lacks the citations. Add sample size calculation citation, Formula name?
5. You should follow the STROBE checklist to report this. Follow the checklist and upload in supplementary file.
6. Why less than 10min time was alloted to study participants?
7. Where is the questionnaire? Attach in supplementary file.
8. Questionnaire contain both open ended and closed ended questionnaire?
9. What are strengths of this study apart of done first time in saudi arabia? What this study adding new in the literature?
10. Still confusing how you evaluated the pharmacist role.
Author Response

(The authors gave the same response as above.)

Round 2
Reviewer 2 Report
None
Author Response
Revisions for the Manuscript ID [healthcare-2108299]-Round 2, entitled “Experience of patients diagnosed with COVID-19 infection, and role of the pharmacist during their infection: the story from the United Arab Emirates” submitted to healthcare Journal.
Dear Editor,
I’m thankful for your consideration of this manuscript.
All the received comments on the manuscript have been taken into account, and I present below the reply to each of them separately.
Academic Editor Notes:
- Following the reviewers' suggestions, the authors have improved their manuscript. Nevertheless, they should format the text following the authors' guidelines (https://www.mdpi.com/journal/healthcare/instructions).
For example:
- Abstract: The abstract should be a total of about 200 words maximum. The abstract should be a single paragraph and should follow the style of structured abstracts, but without headings...;
Done. The abstract was edited as requested
“Abstract: Patients infected with coronavirus have new experiences and hence new needs from the healthcare sector. Acknowledging patients' experiences can exhibit promising outcomes in coronavirus management. Pharmacists are considered a vital pillar in managing patients' experiences during their infection. A cross-sectional study was conducted to assess the experience of COVID-19 infected individuals and the role of pharmacists in the United Arab Emirates. The survey was face and content validated after being developed. Three sections were included in the survey (demographics, experience of infected individuals, and role of pharmacists). Data were analysed using the Statistical Package for the Social Sciences. The study participants (n= 509) had a mean age of 34.50 (SD= 11.93). The most reported symptoms among participants were fatigue (81.5%), fever (76.8%), headache (76.6%), dry cough (74.1%), muscle or joint pain (70.7%), and sore throat (68.6%). Vitamin C was the most used supplement (88.6%), followed by pain relievers (78.2%). Female gender was the only factor associated with symptom severity. About 79.0% agreed that the pharmacist had played an important and effective role during their infection. The most reported symptom was fatigue, with females reporting more severe symptoms. The role of the pharmacist proved to be vital during this pandemic.”
- - References: References must be numbered in order of appearance in the text (including table captions and figure legends) and listed individually at the end of the manuscript.
Done. The references were edited accordingly.
- “Author Contributions” was added according to the guideline
